# Tissue Distribution and Immunomodulation in Channel Catfish (*Ictalurus punctatus*) Following Dietary Exposure to Polychlorinated Biphenyl Aroclors and Food Deprivation

**DOI:** 10.3390/ijerph17041228

**Published:** 2020-02-14

**Authors:** Shannon L. White, Devin A. DeMario, Luke R. Iwanowicz, Vicki S. Blazer, Tyler Wagner

**Affiliations:** 1Pennsylvania Cooperative Fish and Wildlife Research Unit, Pennsylvania State University, 413 Forest Resources Building, University Park, PA 16802, USA; ddemario@fishwildlife.org; 2U.S. Geological Survey, National Fish Health Research Laboratory, Leetown Science Center, Kearneysville, WV 25430, USA; liwanowicz@usgs.gov (L.R.I.); vblazer@usgs.gov (V.S.B.); 3U.S. Geological Survey, Pennsylvania Cooperative Fish and Wildlife Research Unit, Pennsylvania State University, 402 Forest Resources Building, University Park, PA 16802, USA; txw19@psu.edu

**Keywords:** polychlorinated biphenyls, PCBs, Aroclor, PCB homologs, channel catfish, immunomodulation

## Abstract

Although most countries banned manufacturing of polychlorinated biphenyls (PCBs) over 40 years ago, PCBs remain a global concern for wildlife and human health due to high bioaccumulation and biopersistance. PCB uptake mechanisms have been well studied in many taxa; however, less is known about depuration rates and how post-exposure diet can influence PCB concentrations and immune response in fish and wildlife populations. In a controlled laboratory environment, we investigated the influence of subchronic dietary exposure to two PCB Aroclors and food deprivation on tissue-specific concentrations of total PCBs and PCB homologs and innate immune function in channel catfish (*Ictalurus punctatus*). Overall, we found that the concentration of total PCBs and PCB homologs measured in whole body, fillet, and liver tissues declined more slowly in food-deprived fish, with slowest depuration observed in the liver. Additionally, fish that were exposed to PCBs had lower plasma cortisol concentrations, reduced phagocytic oxidative burst activity, and lower cytotoxic activity, suggesting that PCBs can influence stress and immune responses. However, for most measures of immune function, the effects of food deprivation had a larger effect on immune response than did PCB exposure. Taken together, these results suggest that short-term dietary exposure to PCBs can increase toxicity of consumable fish tissues for several weeks, and that PCB mixtures modulate immune and stress responses via multiple pathways. These results may inform development of human consumption advisories and can help predict and understand the influence of PCBs on fish health.

## 1. Introduction

Polychlorinated biphenyls (PCBs) are a group of synthetic chemical compounds that were globally produced and distributed for use in electrical and industrial applications [1]. Due to their uniquely long half-life and concerns about bioaccumulation in the environment, manufacturing of PCBs was banned in most countries the late 1970s and early 1980s [1,2]. Since then, there have been substantial declines in the concentration of PCBs in the environment [3,4]. However, due to bioaccumulation and high biopersistence from slow metabolic dechlorination, PCBs are classified as a persistent organic pollutant and still threaten global wildlife and human health [5]. Despite remediation efforts, PCB concentrations in animal tissues remain at, or above, those detected in previous decades in many historically contaminated ecosystems [6,7,8,9].

PCBs also continue to be released into the environment through improper waste disposal, leaky infrastructure, and burning of industrial materials [10]. Delayed contamination is of particular concern in developing regions, where disturbance of contaminated sediments can remobilize PCBs back into the environment [1]. Further, because PCBs can enter aquatic systems through multiple pathways including direct application, urban and industrial discharges, soil leaching, decay of contaminated tissues, and aerosol and particulate deposition [11,12], riverine environments are particularly susceptible to ongoing PCB contamination. This, along with lipophilic properties of PCBs, makes aquatic organisms, including fishes, mussels, and crayfishes, vulnerable to PCB exposure and contamination—with PCB concentrations in tissues of these taxa up to several orders of magnitude higher than that in the surrounding environment [7,13,14].

Long-term persistence of PCBs in aquatic environments poses significant threats to fish and human health, worldwide. Although PCBs have not been causally linked to declines in fish populations, PCB exposure is associated with acute and chronic immunosuppression [15,16,17] and can increase disease susceptibility [18,19], thereby reducing long-term population fitness and survival. Coplanar PCBs are thought to be the most toxic due to their affinity to the aryl hydrocarbon receptor (AhR), which results in dioxin-like effects via up-regulation of the *CYP1A1* gene [20]. However, recent evidence suggests that non-coplanar PCBs can have endocrine and immunosuppressant effects that are independent of AhR activation [21,22]. Consequently, the PCB mixtures that were widely manufactured and distributed and that contained several different coplanar and non-coplanar PCB congeners can influence immune health via interacting physiological pathways. Previous studies have largely focused on congener-specific immune response, particularly of coplanar PCBs [22], and less is known about the immune response to PCB mixtures.

For humans, PCBs are a probable carcinogen, and consumption of contaminated tissues is associated with birth defects and disruption to endocrine and lymphatic function [23,24]. Given this, many countries have set consumption advisories and/or maximum allowable PCB concentrations in consumable fish and wildlife tissues. For example, in the United States, consumption advisories are issued by state regulatory agencies when the concentration of PCBs in fish tissues exceeds the state’s maximum allowable level of contamination. PCB consumption advisories have resulted in the restriction and/or closure of thousands of global fisheries, which threatens recreational and commercial fisheries and have significant economic and cultural impacts [25,26].

Given concerns about the effects of PCBs on wildlife and human health, there is interest in determining how PCBs assimilate into consumable fish tissues and how exposure can influence long-term fish population demography. However, it is difficult to determine the effects of PCBs when studying wild populations due to the presence of other contaminants and the inability to experimentally control PCB exposure and diet. For example, periods of food deprivation in wild fish stocks have been shown to promote the mobilization of lipids and lipophilic toxicants (including PCBs) from adipose tissue, which can increase tissue toxicity even in the absence of active PCB exposure [27,28,29,30] and lead to prolonged consumption risks. Importantly, as highly chlorinated homologs are less readily metabolized and can concentrate in adipose tissue, secondary exposure to mobilized PCBs may disproportionately increase the concentration of more heavily chlorinated PCB homologs in other tissues [31,32].

While laboratory studies provide the opportunity for more controlled exposure and diet, to date most studies have exposed fish to PCBs using intraperitoneal injections and/or use PCB concentrations that are several orders of magnitude higher than those found in natural systems [33,34,35]. As such, tissue-specific PCB accumulation and immunological response of fish exposed to a diet of environmentally relevant concentrations of PCB mixtures remains poorly understood in commercially important fish species [36].

The objectives of this study were twofold. The first was to determine how subchronic dietary exposure to two common PCB mixtures influenced tissue-specific concentrations of total PCBs and PCB homologs in channel catfish (*Ictalurus punctatus*) and to document immunomodulation immediately following PCB exposure. Channel catfish are an ideal model organism for this study due to their widespread distribution across several continents, and because the U.S. Environmental Protection Agency (EPA) designated channel catfish as a target species for PCB monitoring due to the species’ high rate of recreational harvest and consumption, history of PCB bioaccumulation, and widespread distribution [37]. As such, determining how total PCBs and homolog groups concentrate differentially among tissue types in channel catfish is important for developing sampling protocols and consumption advisories. Additionally, it can also help predict the influence of PCBs on fish health. For example, fillet tissue is most likely to be consumed by humans and the concentration of PCBs in fillet tissue is generally used when setting consumption advisories. However, accumulations in liver and other tissues can be more informative for determining how PCBs influence metabolic, immunological, and endocrine function.

The second goal was to determine how food deprivation altered tissue-specific concentrations of total PCBs and homolog groups and immune response. Wild fish populations often experience periods of limited food availability, which can promote PCB mobilization and influence depuration rates and immune response in natural populations. Therefore, after exposing fish to PCB mixtures, we withheld food for seven weeks while periodically monitoring PCB concentrations and immune function. Understanding the influence of food deprivation is important for management and remediation and can help clarify results from field studies when dietary history is unknown.

## 2. Materials and Methods

This study was conducted at the United States Geological Survey (USGS) Northern Appalachian Research Branch in Wellsboro, Pennsylvania. We used 16,814 L flow-through tank systems that were supplied with water by a nearby spring at a rate of 11 L/min. Water was maintained at 20 °C and dissolved oxygen was maintained above 5 ppm with atmospheric gas delivered via air stones.

The concentration of background PCBs in the spring water influent was measured using semipermeable membrane devices (SPMDs; Environmental Sampling Technologies, Inc., St. Joseph, MO, USA). SPMDs were deployed in a tank with no fish, but that received the same influent as the study tanks. Each SPMD was replaced monthly and tested for total PCBs and homolog groups (see Section 2.4: Tissue Sampling for PCB testing details). The concentration of PCBs in the spring water was negligible throughout the study (average 0.00019 mg/kg ± 0.00011 mg/kg). Moreover, the PCBs that were detected were less-chlorinated homolog biphenyls (i.e., mono, di, and trichloro biphenyls) and were detected only at concentrations that are consistent with ubiquitous background concentrations in natural streams.

### 2.1. Experimental Design

The experimental design outlined here included two phases. During the first phase, herein referred to as the PCB exposure phase, fish were fed either PCB-spiked (P) or unspiked control (C) feed. In the second phase, herein referred to as the food restriction phase, fish were fed either PCB-free (“clean”) food (F) or not fed (NF). Thus, we had four treatments: (1) PCB exposed and clean fed (PF), (2) PCB exposed and not fed (PNF), (3) control fed and clean fed (CF), and (4) control fed and not fed (CNF). This combination of treatments allowed us to fully control for the effects of PCB exposure and food deprivation when comparing responses among treatment groups.

To account for potential effects of fish size, treatments were randomly assigned to tanks of fish using a randomized complete block design with fish size as the blocking variable. Each tank contained 36 randomly selected adult hatchery-sourced channel catfish (total *n* = 576 fish) from Zett’s Fish Farm in Drifting, Pennsylvania. There were four tanks for large fish (average weight = 0.76 kg), eight tanks for intermediate-sized fish (average weight = 0.45 kg), and four tanks of small fish (average weight = 0.22 kg). Thus, each of the four treatments was randomly applied to one tank of large fish, two tanks of intermediate-sized fish, and one tank of small fish, giving four replicates for each treatment.

Fish were sampled on multiple occasions to measure the concentration of PCBs in three tissue types (whole body, fillet, and liver) and immune response. The first sampling event (Time 0; T0) occurred before the PCB exposure phase to document baseline concentrations of PCBs in fish tissues. We did not complete fish health assessments at T0, as fish in the CNF and CF treatments served as adequate controls for determining the effect of PCB exposure on immune system responses. The second sampling event (T1) occurred immediately after the 11-week PCB exposure phase to determine tissue-specific PCB concentrations and immunological response of fish immediately after PCB exposure. The last two samples occurred four weeks (T2) and seven weeks (T3) after the beginning of the food-restriction phase to determine how a prior history of PCB exposure and active food deprivation influenced tissue-specific PCB concentrations and immune response. All testing for total PCBs and PCB homologs was completed using a GC–MS/MS system consisting of an Agilent 6890N GC equipped with a 30 m × 25 mm × 25 µm column coupled to a Waters Autospec Premier MS. Testing was completed by PACE Analytical Laboratories (Minneapolis, MN) in accordance with EPA 1668A guidelines.

All procedures and fish handling were conducted in agreement with the Pennsylvania State University Institutional Animal Care and Use Committee (IACUC, study number 32418).

### 2.2. PCB Exposure Phase

The goal of the PCB exposure phase was to feed fish in the PF and PNF treatments PCB-spiked food until they reached a body burden of 5 mg/kg while feeding fish in the CF and CNF treatments unspiked food. A target concentration of 5 mg/kg was chosen because it is representative of the total body burden found in tissues of wild fish in PCB-contaminated environments [38]. As described below, we intended the PCB exposure phase to last for eight weeks; however, it took 11 weeks for fish in the PF and PNF treatments to reach the target body burden.

Fish were fed commercial-grade pellet feed recommended for large channel catfish. All feed was obtained from the same production number to minimize potential lot-to-lot variation in background PCB contamination. Prior to PCB spiking, three composite feed samples were tested for total PCBs to ensure that background total PCB concentrations were negligible.

PCB-spiked feed was prepared by dissolving a composition of 90% Aroclor 1254 and 10% Aroclor 1260 in fish oil at a concentration of 80 μg/mL total PCB by Accustandard (Newhaven, CT, England). Aroclors 1254 and 1260 were two of the most prevalent Aroclors used in industrial applications and in the environment they often occur at the relative concentrations used in this study. Importantly, these Aroclors are mostly comprised of more highly chlorinated homologs, including both planar and non-coplanar congeners. Subsamples of feed were saturated with PCB-spiked fish oil and had a total PCB concentration of 5.03 ± 0.21 mg/kg, while control feed was saturated in regular, unspiked fish oil and had a total PCB concentration of 0.02 ± 0.0003 mg/kg. Feed was stored in separate, treatment-specific sealed stainless steel containers at 10 °C.

We periodically sampled one fish from each of the PF and PNF tanks to monitor increases in PCB body burden. To maintain constant stocking densities across tanks, we also removed one randomly selected fish from each of the CF and CNF tanks. Five weeks into the PCB exposure phase we determined that the concentration of PCBs in the spiked feed was not high enough to reach the desired 5 mg/kg body burden in a reasonable timeframe. Therefore, the concentration of PCBs in the fish oil was increased to 720 μg/mL (while maintaining 90% Aroclor 1254 and 10% Aroclor 1260) to reach mean total PCB concentrations in spiked feed of 11.7 ± 0.25 mg/kg. After 11 weeks, body burden of PN and PNF fish reached approximately 5 mg/kg, and T1 samples were collected from all tanks.

### 2.3. Food Deprivation Phase

At the conclusion of the PCB exposure phase, fish in the PNF and CNF treatments were food deprived, and fish in the PF and CF treatments were fed non-spiked feed that had not been saturated in fish oil. The food deprivation phase lasted for seven weeks, which was deemed an appropriate amount of time for lipid mobilization while avoiding serious adverse health effects for fish in the no-food treatments. Tissue samples were taken four weeks (T2) and seven weeks (T3) after the start of the food deprivation phase.

### 2.4. Tissue Sampling 

Four fish were randomly sampled from each tank during each sampling event (T0, T1, T2, T3) and euthanized with a lethal dose of MS-222. Prior to tissue sampling we withheld food for 48 h to ensure there was no residual undigested feed in the digestive system that could bias our measurements of PCB concentrations. For each sampled fish, we measured fork length and body weight and examined fish for superficial external abnormalities. Using a heparinized 5-cc syringe, we drew blood from the caudal vasculature and transferred samples to a heparinized vacutainer. All samples were collected within 3 min of capture and placed on wet ice before separating the cellular fraction of the blood from plasma by centrifugation. Plasma was removed, transferred to cryovials, and frozen on dry ice.

After collecting blood samples, we preserved one fish for total body burden analysis. For the remaining three fish, we aseptically dissected the anterior kidney, gonads, liver, and right fillet. The anterior kidney tissue was processed as described by [19]. Briefly, anterior kidney tissue was aseptically removed and placed into processing medium (PM; L-15 medium supplemented with 2% fetal bovine serum (FBS), 100 U mL^−1^ penicillin, 100 µg mL^−1^ streptomycin, and 10 U mL^−1^ sodium heparin) and stored on wet ice prior to cell processing. We weighed the gonads and calculated the hepatosomatic index (HSI) for all sexually mature individuals. HSI was calculated as the liver weight/(total body weight − gonad weight) × 100 [39,40]. We packaged fillet and portions of the liver for PCB congener analysis. To ensure that tissue samples were of sufficient mass for analysis, a composite tissue sample from three individual fish per tank was collected at each sample event. Preserved tissue pieces were routinely processed, embedded into paraffin blocks, sectioned at 5 µm and stained with hematoxylin and eosin and Perl’s Prussian blue iron stains to examine possible microscopic pathologies.

### 2.5. Bactericidal Activity

Assessment of adherent anterior kidney leukocytes (AKLs) ability to kill the salmonid pathogen Yersinia ruckeri was determined using the method described by [41]. In short, 2 × 10^6^ leukocytes suspended in AM were added in quadruplicate to the wells of a 96-well plate for bacterial challenge and in triplicate on the same plate for subsequent adherent cell enumeration. Leukocytes from fish from all treatments were included on all plates to minimize the effect of inter-plate variability.

After incubating plates at 20 °C for 2 h, we replaced the media in all wells with culture media, and then cultured plates at 20 °C in a humidified chamber for 36 h to allow activated leukocytes to reach a resting state. We then removed the culture media from all wells, washed wells with antibiotic-free unsupplemented L-15 medium, and added 100 μL of antibiotic-free supplemented L-15 with 5% FBS. A 48 h culture of *Y. ruckeri* (Hagerman strain; NFHRL # 11.40) washed and suspended in Hanks’ balanced salt solution (HBSS; OD_600_ = 0.5) was added to the treatment wells and to a row of cell-free control wells in a volume of 25 μL. The plates were incubated in a humidified chamber at 20 °C for 4 h. Media was subsequently removed from the treatment and control wells, cells were lysed with lysis buffer (0.2% Tween 20 in dH_2_O), and the lysate was immediately serially diluted in serocluster plates containing tryptic soy broth. Diluted lysates were plated onto tryptic soy agar plates, and we counted the number of colony forming units (CFUs). Bactericidal activity was expressed as % CFU reduction (1 − (CFU treated/CFU control) × 100) where CFU treated = mean CFU value for replicate wells with adherent leukocytes and CFU control = mean CFU value for replicate wells with media only. A corrected % CFU reduction was defined as % CFU reduction x cell density correction, where cell density correction = mean number of adherent cells from the same cell source used to determine bactericidal activity/1 × 10^6^ cells.

### 2.6. Respiratory Burst Activity

The production of reactive oxygen species (ROS) from anterior kidney samples was determined using the nitroblue tetrazolium (NBT) method [42]. In short, 1 × 10^6^ leukocytes suspended in AM were added to the wells of a clear 96-well plate, incubated for adherence, washed, and subsequently incubated in CM at 20 °C for 48 h to allow cells to reach a resting state. Culture medium was removed, and cells were gently washed with room-temperature HBSS. Cells were then activated with 50 ng/mL of PMA in CM containing 1 mg/mL NBT, and control cells were incubated in the presence of 1 mg/mL NBT without PMA. Cells were incubated at 20 °C for 30 min and then the supernatant was removed and cells were washed with HBSS to remove unreduced NBT. Reduced NBT was solubilized with the sequential addition of 60 µL of 2M KOH and 70 µL of DMSO followed by rigorous pipetting. A volume of 100 µL of solubilized NBT was transferred to a new multiwell plate and absorbance was measured at 600 nm. We calculated the stimulation index (SI) as the replicate mean OD_600_ for a given set of PMA-stimulated leukocytes divided by the replicate mean OD_600_ of the associated PMA-free control leukocytes.

### 2.7. Cytotoxic-Cell Activity 

The ability of channel catfish anterior kidney leukocytes to lyse virally transformed epithelioma papulosum cyprini (EPC) cells was determined using the calcein AM release-based cytotoxic cell assay as described by [43]. Briefly, an 18 h culture of EPC cells plated at a density of 1 × 10^5^ cells well^−1^ in 96-well plates were incubated with 5 µM calcein AM in CM for 5 h. Cells were then washed four times with room-temperature DPBS and cultured in CM for an additional 30 min. Media was removed and effector AKLs suspended in CM were added in quadruplicate at effector to target ratios of 10:1, 2:1 and 1:1. Lysis buffer (25 mM sodium borate, 0.1% Triton-X100 in CM, pH 9.0) or CM alone was added in quadruplicate to another set of wells to determine total and spontaneous release, respectively. Plates were centrifuged at 50 g for 5 min and incubated at 20 °C for 8 h in the dark. A volume of 50 µL supernatant was removed from all wells and added to a 96-well black plate pre-loaded with 2× lysis buffer. Supernatant from the total release wells were added to 1× lysis buffer. Fluorescence intensity (FI) was measured by reading the plates from the bottom using a SpectraFluorPlus (Ex = 485, Em = 535 and gain = 60). We quantified percent cytotoxic cell activity as (Experimental release − Spontaneous release)/(Total release − Spontaneous release) × 100.

### 2.8. Plasma Cortisol 

Plasma cortisol was determined using a competitive enzyme-linked immunoassay modified from [44]. Adaptations of this assay included the sourcing of cortisol-HRP-conjugate and rabbit anti-cortisol antibody from Fitzgerald Industries (Acton, MA, USA). In addition, SureBlue Select (KPL Inc.; Seracare, Milford, MA, USA) was used as the enzyme substrate for color development. Optical density was read using a V_max_ Kinetic Microplate Reader (Molecular Devices, Sunnyvale, CA, USA), and plasma cortisol concentrations were determined via interpolation to a 4-parameter standard curve. The range of the assay defined by the standard curve was 1–400 ng/mL.

### 2.9. Statistical Analysis

We used a Bayesian mixed-effects ANOVA to compare tissue-specific concentrations of total PCB and homolog groups and measures of immune function among the four treatments and across the four sample times. The final means-parameterized model, which was used for all response variables (total PCB concentrations, homolog concentrations, body weight, HSI, blood plasma cortisol concentrations, stimulation index, bactericidal activity, and cytotoxic cell activity) was as follows:(1)$$y_{i}\sim N\left(\alpha_{kl\left(i\right)}\times {\mathrm{treatment}}_{i}\times {\mathrm{time}}_{i}+\gamma_{j\left(i\right)},\sigma_{k}^{2}\right),\;for\;i=1\ldots n\;\mathrm{observations}$$
$$\gamma_{j}\sim N\left(0,\tau^{2}\right),\;for\;j=1\ldots J\;\tan\mathrm{ks}$$
where *y_i_* is the response for observation *i* and $\alpha_{kl}$ is the mean response for treatment k at time *l*. The model included the fixed effects of time (three or four sampling events, depending on the response variable: T0, T1, T2, T3), treatment (CF, CNF, PF, PNF), and a time x treatment interaction. A random effect for tank *j*, $\gamma_{j}$, was included to account for subsampling from each tank and was assumed to be independent and identically distributed with a mean of 0 and variance $\tau^{2}$. Response variables were log_e_ transformed prior to analysis to accommodate the assumption of normality. Additionally, to accommodate the assumption of homogeneity of variance, a separate residual variance parameter was estimated for each treatment, *k* (i.e., $\sigma_{k}^{2}$).

Uninformative normal priors were used for $\alpha_{kl}$ and uninformative uniform priors were used for $\sigma_{k}$ and τ. For each response variable, we ran three parallel Markov chains, beginning each chain at a different random starting value. Each chain was run for 10,000 iterations, from which the first 5000 samples were discarded, resulting in 15,000 samples used to summarize the posterior distributions. Model convergence was assessed using the scale reduction factor ($\hat{R}$) for each parameter, trace plots, and plots of posterior distributions. Significant differences between treatment groups and sampling times were determined based on whether or not the 90% credible interval of the difference between the groups and/or times being compared overlapped with zero. Unless otherwise noted, all values are presented as posterior means with 90% Bayesian credible intervals. All statistical analyses were completed in JAGS in the jagsUI package [45] using R statistical software 3.6.0 (R Core Team, Vienna, Austria [46]). For ease of interpretation, response variables (and posterior distributions) that were modeled on the log_e_ scale were back-transformed when reporting results.

## 3. Results

### 3.1. Fish Weight and HSI

PCB exposure did not influence fish weight as, across all sample events, the posterior mean weights of fish in the PF and PNF treatments were similar to the posterior mean weights of fish in the CF and CNF treatments, respectively. At the end of the food restriction phase (T3) the posterior mean weights of fish in the PNF and CNF treatments were lower than the posterior mean weights of fish in the PF and CF treatments; however, the effect was not statistically significant (Figure 1A).

Subchronic PCB exposure had no immediate effect on HSI, as fish in all treatments had a similar HSI at T1 (posterior mean HSI (and 90% CI) at T1 for CF = 1.51 (1.37, 1.66), CNF = 1.59 (1.47, 1.72), PF = 1.37 (1.20, 1.56), PNF = 1.63 (1.50, 1.77). However, food deprivation significantly decreased HSI, with the strongest effect observed at T3. Comparing fish with the same history of PCB exposure, the posterior mean differences in HSI between fish in the CF and CNF treatments = 0.79 (0.63, 0.93) and PF and PNF treatments = 0.81 (0.58, 0.99). Conversely, posterior mean HSI was similar in fish in the CNF and PNF treatments (posterior mean difference = 0.01 (−0.08, 0.09) and CF and PF treatments (posterior mean difference = 0.003 (−0.27, 0.21), suggesting that food deprivation, and not PCB exposure, had the larger effect on HSI.

### 3.2. Microscopic Pathology

Microscopic examination of liver tissue at sample times T2 and T3 showed reduction of both glycogen and lipids in hepatocytes of CNF and PNF. Hepatocytes of fed catfish were highly vacuolated and there were few to no macrophage aggregates within the hepatic tissue. Hepatocytes in livers from the CNF and PNF groups were reduced in size and showed a lack of vacuolization. In the PNF group, many of the livers also had areas of darkly staining apoptotic and regenerative cells and small accumulations of ceroid/lipofuscin-containing cells. One liver from this group contained a vacuolated preneoplastic focus (Appendix A).

### 3.3. Total PCB Concentrations

Total PCB concentrations in whole fish, fillet, and liver tissues for fish in the CN and CNF treatments did not exceed baseline measurements at any sample point (i.e., concentrations did not exceed 0.07 mg/kg). By the end of the PCB exposure phase (T1), posterior mean whole body total PCB concentrations for fish in the PF and PNF treatments increased to 4.11 mg/kg (3.04, 5.40) and 5.24 mg/kg (3.87, 6.88), respectively. There was no significant change in whole body PCB concentrations by T2. However, by T3, whole body total PCB concentrations declined by 47% to 2.16 mg/kg (1.59, 2.84) in fish from the PF treatment, whereas whole fish total PCB concentrations in fish from the PNF treatment declined by only 27% to 3.79 mg/kg (2.79, 4.96). Notably, at this time, PCB depuration was faster for fish in the PF treatment (estimated difference in total PCB concentrations in whole fish from the PNF and PF treatments at sampling point T3 = −1.63 mg/kg [−2.93, −0.72; Figure 2A]).

Posterior mean total PCB concentrations in fillet tissue from fish in the PF and PNF treatments increased to 1.82 mg/kg (1.50, 3.23) and 2.22 mg/kg (2.17, 4.32), respectively, at T1. However, unlike whole body samples, total PCB concentrations in the fillet were similar between fish in the PF and PNF treatments at all sample points. Further, we did not observe a significant decline in total PCBs in the fillet through time, as PCB concentrations decreased by only 25% and 21% in the PF and PNF treatments, respectively (Figure 2B).

Food deprivation significantly influenced the rate of total PCB depuration in the liver. For fish in the PF treatment, the posterior mean total PCB concentration in the liver was 0.32 mg/kg (0.18, 0.49) at T1, and declined by 46% to 0.17 mg/kg (0.13, 0.35) by T3 (estimated difference in total PCB concentration in liver tissue of fish in the PF treatment at time T1 and T3 = 0.14 mg/kg (0.06, 0.22). In contrast, the posterior mean total PCB concentration in liver tissue for fish in the PNF treatment reached 0.34 mg/kg (0.18, 0.58) after PCB exposure, and then increased by 27% to 0.43 mg/kg (0.23, 0.73) by T3. While this increase was not statistically significant, it is in contrast to the significant declines in total PCBs found in the liver of fish in the PF treatment (Figure 2C).

Across tissue types, posterior mean total PCB concentrations were significantly lower in the liver compared to the fillet and whole body for fish in both the PF and PNF treatments and across all sample periods. The fillet also had significantly lower posterior mean total PCB concentrations until T3, at which point faster depuration in the whole body resulted in no statistically significant difference in PCB concentrations between the whole body and fillet.

### 3.4. Homolog Groups

Posterior mean concentrations of each of the ten homolog groups (mono-through decachlorobiphenyl) generally followed similar tissue-specific patterns of depuration as those described for total PCB concentrations. Specifically, fish from the PF treatment had faster rates of depuration than fish in the PNF treatment and, for most homolog groups, there was a decline in posterior mean PCB concentrations in whole body and fillet tissues by T3. However, declines in PCB concentrations were significant only for less-chlorinated homolog groups. As with total PCBs, by T3 posterior mean PCB concentrations in the liver increased in PNF fish, but decreased in PF fish, particularly for lower-chlorinated homolog groups (Appendix A).

### 3.5. Functional Immune Responses and Stress Physiology

At T1, plasma cortisol concentrations were marginally lower in fish from the PF and PNF treatments; however, the effect was not statistically significant (posterior mean plasma cortisol in CF = 9.13 ng/L (7.97, 10.37), CNF = 9.57 ng/L (7.81, 11.47), PF = 7.22 ng/L (6.13, 8.41), PNF = 7.71 ng/L (5.96, 9.74). Food deprivation significantly influenced plasma cortisol concentrations, but the direction of the effect was dependent on prior PCB exposure. Specifically, while posterior mean plasma cortisol concentrations in fish from the CNF treatment significantly increased by T3 to 19.94 ng/L (14.73, 21.63) plasma cortisol concentrations for fish in the PNF treatment significantly decreased to 4.96 ng/L (3.84, 6.26). Plasma cortisol concentrations remained stable for fish in the CF and PF treatments (Figure 3A).

PCB exposure led to reduced oxidative burst activity, as fish from the PF and PNF treatments had significantly lower oxidative burst activity than fish in the CF and CNF treatments at T1 (posterior mean stimulation index in CF = 2.00 [1.72, 2.30], CNF = 1.92 [1.71, 2.17], PF = 1.49 [1.27, 1.74], PNF = 1.45 [1.26, 1.65]). By T2, fish from all treatments had similar oxidative burst activity, and by T3, food-deprived fish had a significantly lower stimulation index than fish consuming PCB-free feed, regardless of PCB exposure history (posterior mean stimulation index at T3 in CF = 2.40 [2.08, 2.77], CNF = 1.17 [1.04, 1.32], PF = 2.18 [1.86, 2.55], PNF = 1.22 [1.06, 1.39]). Compared to fish from the CF treatment at T1, this represents a 34% and 30% decline in the stimulation index in fish from the CNF and PNF treatments, respectively (Figure 3B).

At T1, posterior mean bactericidal activity was significantly lower in fish from the PF treatment compared to fish from the CF treatment (posterior mean bactericidal activity in CF = 35.65 [29.07, 43.29], PF = 19.05 [14.11, 24.92]). However, while fish from the PNF treatment had a lower bactericidal activity than fish from the CNF treatment, the effect was not statistically significant suggesting an overall weak effect of PCB exposure on bactericidal activity. Similar to oxidative burst activity, irrespective to PCB exposure history, food deprivation resulted in a significant decline in bactericidal activity by T3 (posterior mean bactericidal activity in CF = 35.53 [28.81, 43.00], CNF = 0.95 [0.24, 2.32], PF = 31.66 [23.39, 41.34], PNF = 1.39 [0.40, 3.24]). Compared to the posterior mean bactericidal activity at T1 from fish from the CN treatment, by T3 bactericidal activity declined by 93% and 91% in fish from the CNF and PNF treatments, respectively. Conversely, bactericidal activity significantly increased in PF fish by T3 (estimated difference in bactericidal activity from fish in the PF treatment at sampling points T1 and T3 = 12.61 mg/kg (3.44, 22.91; Figure 3C), suggesting that PCB exposure did not have long-term effect on bactericidal activity and that recovery from exposure is possible.

Cytotoxic cell activity was similar across all four treatment groups at T1 (posterior mean cytotoxic cell activity in CF = 15.51 [13.46, 17.78], CNF = 17.64 [11.34, 25.89], PF = 15.29 [11.66, 19.45], PNF = 16.79 [11.95, 22.75]). Cytotoxic cell activity started declining at T2 for fish from the CNF, PF, and PNF treatments, and by T3 fish from these treatments had significantly lower cytotoxic cell activity than fish from the CF treatment (posterior mean cytotoxic cell activity at T3 in CF = 17.33 [15.02, 19.86], CNF = 4.62 [2.96, 6.76], PF = 11.00 [8.38, 14.11], PNF = 3.45 [2.44, 4.67]). Compared to fish from the CF treatment at T1, cytotoxic cell activity declined by 56% and 70% for fish from the CNF and PNF treatments, respectively, whereas cytotoxic cell activity declined by only 28% for fish from the PF treatment. Together, these results suggest that PCB exposure and food deprivation both decrease cytotoxic cell activity, with food deprivation having the stronger effect.

## 4. Discussion

This is one of the first studies to investigate the combined effects of dietary exposure to PCB mixtures and food deprivation on tissue-specific concentrations of total PCBs and homolog groups and functional immune response in a controlled setting. Similar to results of other studies, we observed no effect of PCB exposure alone on fish weight or hepatosomatic index [47,48]. However, we found that the concentration of total PCBs and homolog groups in the whole body, fillet, and liver can remain elevated for at least seven weeks post-exposure, and that the rate of PCB depuration is tissue-specific and dependent on nutritional status. In particular, we observed an increase in PCB concentrations in the liver of food-deprived fish, suggesting that the mobilization of PCB-rich lipids can have delayed effects on tissue toxicity [49]. PCB exposure did have immunomodulatory effects; however, the effect of food deprivation was generally stronger for the parameters evaluated.

While PCB uptake mechanisms are well understood for many aquatic species, less is known about the timing and pathways of PCB elimination [50]. Thus, while it is often possible to estimate PCB concentrations in the abiotic environment, without repeated sampling of animal tissues it can be difficult to assess the speed and success of full ecosystem recovery [51]. Tissue sampling can be financially and logistically difficult, and delays in removing consumption restrictions on waterways can have significant effects to commercial and recreational fisheries. Accordingly, there is interest in developing predictive models that can better estimate tissue bioaccumulation and toxicity that can assist in the development of more realistic ecosystem recovery goals and objectives [25]. We show that short-term dietary exposure to environmentally relevant concentrations can increase tissue PCB concentrations for several weeks, particularly during periods of limited food availability. This result suggests that even short-term exposure to PCBs, which may be expected after damage to riparian infrastructure or in individuals that temporarily occupy a contaminated river reach, can significantly influence fish and human health for several weeks after exposure. This finding also suggests that ecosystems with long-lived and/or migratory species, which could be exposed to variable concentrations of PCBs, may remain contaminated despite low concentrations of PCBs in the abiotic environment.

Understanding how PCB exposure influences specific tissue bioaccumulation and immune responses is difficult in natural environments due, in part, to the inability to control forage quality and quantity. Similar to [52], we show that diet can significantly influence PCB concentrations in fish tissues and modulate the effect of PCB exposure on immune response. Accordingly, studies that do not account for recent diet and nutritional status should be viewed cautiously, as they can provide incomplete information about the extent of ecosystem recovery and may be inappropriate for informing human consumption advisories [18]. For example, while the difference in response between fish in the PF and PNF treatments was not always statistically significant, depuration rates were up to two times faster for fish in the PF treatment. This indicates that adequate forage after PCB exposure can lead to substantial declines in total body burden that are not seen in food-deprived fish. Fish often experience seasonal food deprivation, particularly during winter or ontogenetic migrations. Thus, additional studies that consider species-specific forage ecology and migration could be beneficial for predicting the risk of PCB exposure [53]. Additionally, while we fed fish a constant diet of PCBs, a more realistic exposure scenario that alternates between forage with variable levels of PCB toxicity may provide more realistic conclusions about tissue depuration rates.

It is important to consider that, due to animal welfare concerns, the duration of the food deprivation phase here was limited to seven weeks. While this design limits our ability to make inferences about the long-term effects of PCB exposure and food deprivation, our results still provide insights into how PCBs may influence fish and human health through consumption of contaminated fish tissues. For example, we found that PCB concentrations in the fillet, which is the tissue most likely to be consumed by humans, may depurate more slowly than concentrations in the whole fish. This information is important for establishing human consumption advisories and monitoring protocols and for understanding how PCBs bioaccumulate across trophic levels.

Similarly, our results can help elucidate the immunological response of channel catfish following PCB exposure and food deprivation. While the effect of food deprivation was generally greater than that for PCB exposure, PCB exposure did decrease bactericidal and oxidative burst activity. This result is consistent with those of previous studies in ictalurids [19,38,54], and suggests that PCB exposure may increase disease susceptibility, which could influence population demography. Notably, bactericidal activity returned to normal seven weeks after the start of a PCB-free diet, suggesting that the effect of PCBs on some immunological responses may be temporary. Overall, the rapid return of immune function following the initiation of a PCB-free diet could be because kidney and liver tissues, which have high metabolic activity, depurated PCBs to concentrations that were no longer biologically relevant, which resulted in no significant immunomodulation.

While functional immune responses changed most significantly to food deprivation, the effects of PCB exposure and food deprivation appear to be additive for plasma cortisol. Fish in the PF treatment had marginally lower plasma cortisol concentrations than fish in the CF treatment at all sampling times, suggesting that PCB exposure can decrease cortisol concentrations even after several weeks of consuming a PCB-free diet. Moreover, by T3, fish in the PNF treatment had significantly lower plasma cortisol concentrations than fish in the other treatments. The most notable difference in cortisol at T3 was between fish in the PNF and CNF treatments, highlighting the effect of PCB exposure independent of food deprivation. This result is consistent with those of other studies, which found that fish sampled from contaminated environments lacked the ability to increase serum cortisol concentration in response to acute stress [55,56], perhaps due to impaired steroidogenesis [57]. While we did not analyze the pituitary microscopically, other studies have found atrophic pituitary corticotropes in fish exposed to PCBs. This finding suggests that PCB exposure likely attenuates the cortisol response by way of hypothalamus–pituitary–interrenal (HPI) axis disruption [58].

Overall, decreased immune function associated with food deprivation is not surprising, and may be associated with inadequate energy reserves for metabolically depending immunocyte responses [59], as well as the combined influence of food deprivation and PCBs on oxidative stress [60]. Both fish in the CNF and PNF treatments had evidence of reduced lipid and glycogen energy storage within hepatocytes. The PNF group also had accumulations of ceroid/lipofuscin, which is associated with oxidative damage [61]. There remain significant uncertainties as to the extent that PCBs may interrupt HPI function, and future studies could incorporate additional analyses of hormone production such as thyroxine and sex steroid hormones.

It is challenging to determine how PCB-induced tissue toxicity and immunosuppression translate into population- and ecosystem-level effects. Aquatic organisms are rarely exposed to a single toxicant, nor are the exposure mechanisms and concentrations well understood. Additionally, environmental and biological stressors, such as competition, predation, and environmental stochasticity, can work synergistically to further suppress immune function, with the strength of the synergism dependent on species-specific ecology [62]. For example, environmental context may exert a strong control on fish response to PCBs in salmonid species, which have a relatively narrow temperature tolerance and have strong intraspecific competition for resources. Additionally, PCBs can have chronic effects on metabolic capacity which may be realized only during periods of prolonged starvation and/or ontogenetic changes in behavior and resource use [49]. Understanding these interactive effects, along with the potential for local adaptation to stress [63], will be important for predicting how PCB exposure, and contaminant exposure in general, influences population vulnerability to disease and disturbance, and for providing more accurate timelines of ecosystem recovery following PCB contamination [64].

Given the ubiquity of PCBs in aquatic environments and the occurrences of fish kills worldwide, the development of species-specific depuration models and robust biomarkers for PCB toxicity would be advantageous for assessing consumption risk and for developing remediation timelines [65]. Our controlled study provided insights into tissue-specific toxicity and immune response in channel catfish exposed to a diet with PCBs, and potential long-term effects of PCB exposure after an acute period of food deprivation. While additional studies investigating the influence of other abiotic and biotic stressors are warranted, results shown here can be used to help clarify field-based studies and provide evidence that PCB exposure does influence innate immune response and can result in long-term tissue toxicity. This, and similar studies, are important for understanding present-day PCB depuration rates and for predicting future changes in PCB exposure with environmental change.

## 5. Conclusions

These results suggest that short-term dietary exposure to PCBs can increase toxicity of consumable fish tissues for several weeks, and that PCB mixtures modulate immune and stress responses in fishes via multiple interacting pathways. While PCBs can quickly depurate below the concentration that warrants human consumption advisories, lack of adequate nutrition can delay depuration and prolong human exposure risks. Thus, it may be important to consider species’ ecology and food availability when assessing human health concerns associated with the consumption of fishes from PCB-contaminated ecosystems.

Overall, these results can be informative in the development of global fish consumption advisories and can also be used to predict the risk of PCBs to human health. Additionally, our findings of immunomodulation following PCB exposure and food deprivation can help elucidate how PCBs influence fish and wildlife population demography.

## Figures and Tables

**Figure 1 ijerph-17-01228-f001:**
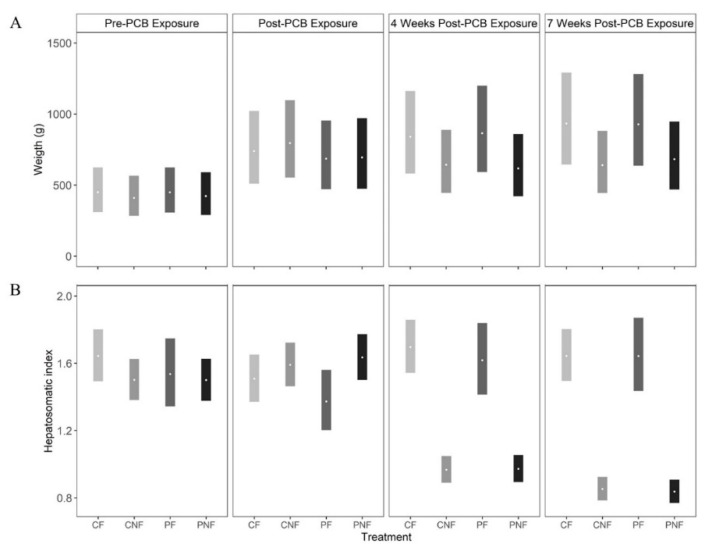
Posterior mean (open circles) and 90% credible intervals (bars) for (**A**) weight and (**B**) hepatosomatic index for channel catfish in each of four treatments (CF = control and fed, CNF = control not fed, PF = polychlorinated biphenyl (PCB) exposed and fed, PNF = PCB exposed and not fed). Weight and hepatosomatic index were measured before PCB exposure, after 11 weeks of PCB exposure, and four and seven weeks after PCB exposure, during which time fish in the CNF and PNF treatments were also deprived of food.

**Figure 2 ijerph-17-01228-f002:**
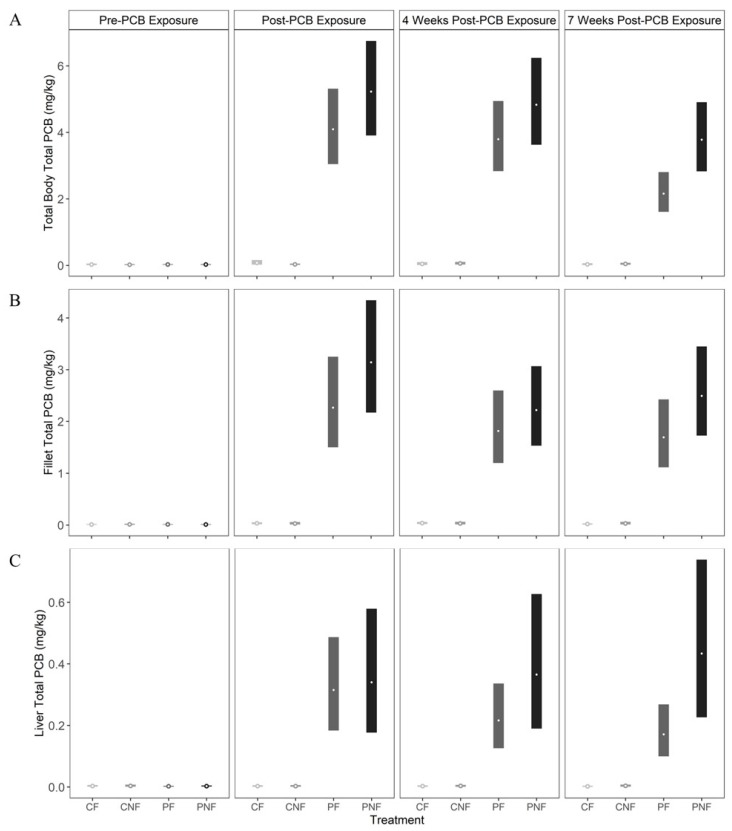
Posterior mean (open circles) and 90% credible intervals (bars) for total PCB concentrations in (**A**) whole body, (**B**) fillet, and (**C**) liver tissues measured for channel catfish in each of four treatments (CF = control and fed, CNF = control not fed, PF = PCB exposed and fed, PNF = PCB exposed and not fed). PCB concentrations were measured before PCB exposure, after 11 weeks of PCB exposure, and four and seven weeks after PCB exposure, during which time fish in the CNF and PNF treatments were also deprived of food.

**Figure 3 ijerph-17-01228-f003:**
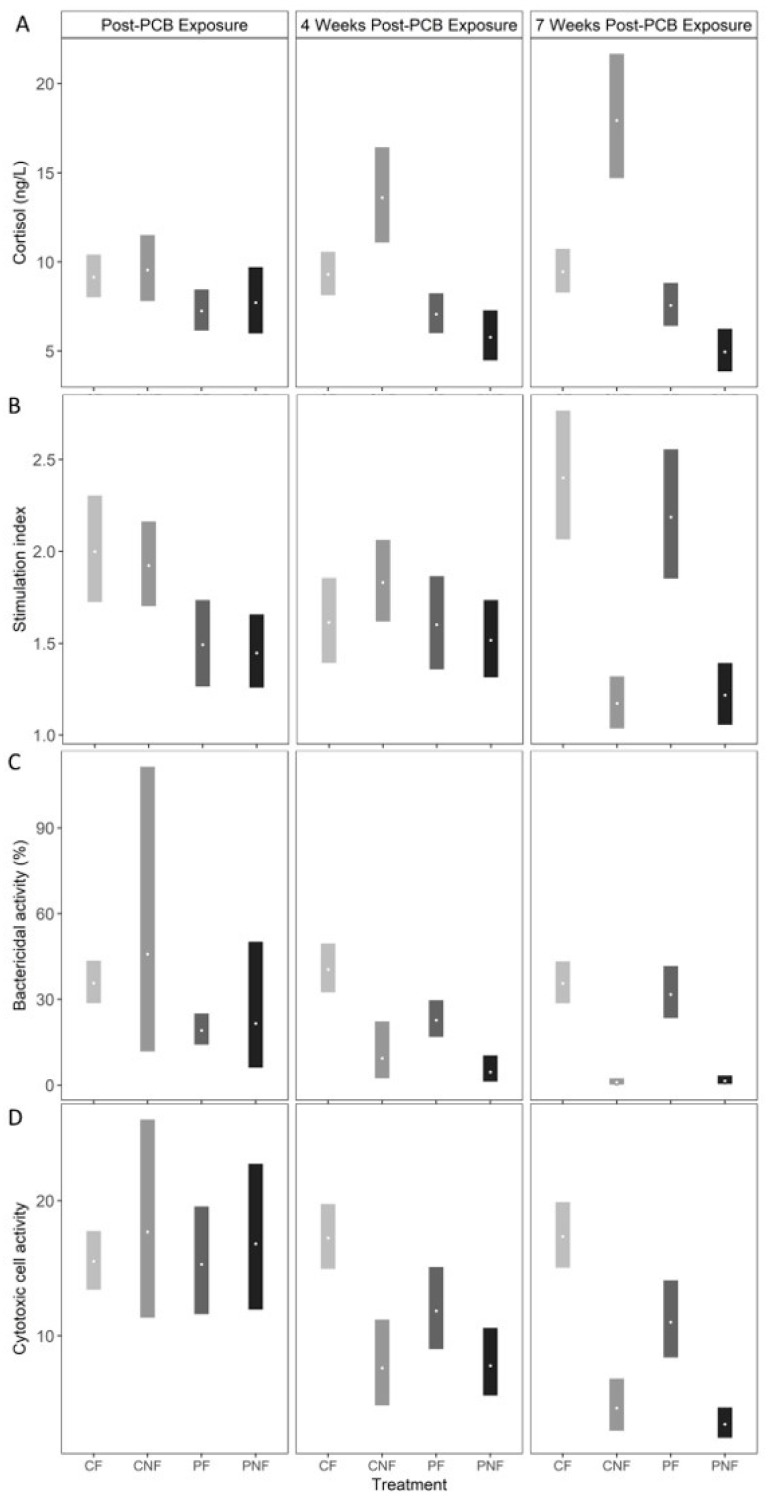
Posterior mean (open circles) and 90% credible intervals (bars) for (**A**) blood plasma cortisol, (**B**) phagocytic burst activity index, (**C**) bactericidal activity, and (**D**) cytotoxic cell activity measured for channel catfish in each of four treatments (CF = control and fed, CNF = control not fed, PF = PCB exposed and fed, PNF = PCB exposed and not fed). PCB concentrations were measured after 11 weeks of PCB exposure and four and seven weeks after PCB exposure during which time fish in the CNF and PNF treatments were also deprived of food.

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
