# Peer review of "Tissue Distribution and Immunomodulation in Channel Catfish (Ictalurus punctatus) Following Dietary Exposure to Polychlorinated Biphenyl Aroclors and Food Deprivation"

_ijerph, 2020, doi:10.3390/ijerph17041228_

Round 1
Reviewer 1 Report
While PCBs were banned in most countries decades ago, they persist in aquatic ecosystems and can bioaccumulate up food chains, ultimately posing risks to aquatic life and human consumers. Improved understanding of effects of PCBs on the host would inform risk assessment, and of how PCBS are distributed among fish tissue would inform consumption advisories. In a laboratory study, White et al. measured the tissue distribution of two arlochlors among tissues of channel catfish, a widely consumed species, and showed how the presence of these compounds affected immune response of the fish. The experimental design and analyses are appropriate, and the results will prove of interest to a range of readers in the environmental risk assessment and fish immunology communities. My comments are mostly aimed at improving clarity of presentation. I make some comments here and also provide a marked manuscript – there are many small, context-specific grammatical glitches.
Introduction. – At line 39, the authors might provide a citation supporting the important statement of how PCBs were used.
At line 140, would PCBs be manufactured UNintentionally?
At line 74, the authors might cite an example or two of U.S. states having issued consumption advisories. More importantly, a non-sequitor follows. Would state consumption advisories lead to closure of GLOBAL fisheries? There is no linkage; a transitional statement is needed, something along the lines of: Bans at the national level have resulted in the restriction or closure…
At line 98, channel catfish is native to eastern North America. While it has been widely distributed, it is not globally so, and the statement should be qualified.
At line 109, the authors might state that food limitation is an important aspect of fish feeding ecology, thereby explaining why feed deprivation might be of interest in a study of PCB mobilization.
Methods. – At line 164, the authors should report the manufacturer of the feeds used.
Results. – I’ve marked a number of small grammatical errors on the manuscript.
Discussion. – At line 443, the authors might also mention that the findings can assist in development of fish consumption advisories, a key point that is not mentioned.
At line 449, it’s not ecosystems, but rather fish that should be the subject of this sentence.
At line 474, the authors might call for a study of PCB transfer up aquatic food chains.
References. – I’ve marked a few minor glitches in the literature citations.

Reviewer 2 Report
Suggest adding in some more details about PCB concentration analysis. While stating the EPA guideline was followed is good it would be worthwhile to include specifics about actual instruments and columns used for this particular analysis.
Consider adding pathology data at least in supplemental information. Results reported are fine but showing some actual data would be useful to support what you say.
Be sure to refer to where data is located in figures. There are a couple places where this is not done.
